# First record of the smallest extant brachiopod *Gwynia capsula* (Jeffreys, 1859) from the German Bight

Carsten Lüter[1,2]*, Anke Sänger[2], Fabia Wagner[2]

**1** Humboldt-Universität zu Berlin, Institut für Biologie, Unter den Linden 6, Berlin, Germany, **2** Museum für Naturkunde, Leibniz-Institut für Evolutions- und Biodiversitätsforschung, Invalidenstr. 43, Berlin, Germany

* carsten.lueter@mfn.berlin

## Abstract

We describe for the first time the occurrence of the smallest extant brachiopod, *Gwynia capsula* (Jeffreys, 1859), from the island of Helgoland. A single living specimen was found in dredged shell gravel from the Helgoland trench ("Tiefe Rinne") mainly consisting of large dead shells of the bivalves *Ostrea edulis* and *Modiolus modiolus*. *G. capsula* was identified through its minute size and its characteristic submarginal ridges on the inside of the dorsal valve supporting the trocholophous lophophore of the animal. Among other localities, populations of *G. capsula* are known from British waters as well as the continental coasts of, e.g., France, Belgium and the Netherlands. However, the reproductive biology of the species makes it rather unlikely that larvae of *G. capsula* have reached Helgoland by natural drift. It is briefly discussed whether ship-based trading throughout the 19th century may have had an influence here.

## Introduction

Of about 400 described extant species of Brachiopoda only a few have been recorded from the North Sea, most of which from British coasts. Atlantic water masses enter the North Sea at the Shetland and Orkney Islands and run counter-clockwise through its basin causing an amphidromic point in the centre. These water masses may act as a means for dispersal of brachiopod larvae from British populations into the basin. However, since these larvae are lecithotrophic (non-feeding) and therefore only short-lived, the German Bight situated at the southeastern-most corner of the North Sea basin is rather unlikely to receive sufficient numbers of brachiopod larvae for establishing stable populations. Additionally, all brachiopod species in the North Sea are hard bottom dwellers and the vast muddy sand flats of the wadden sea are unsuitable settling grounds for this animal group. The only exception to this situation is the island of Helgoland (Fig 1), a large sandstone rock about 50 km off the

**Data availability statement:** All relevant data are within the manuscript.

**Funding:** The author(s) received no specific funding for this work.

**Competing interests:** The authors have declared that no competing interests exist.

German coastline. Accordingly, Helgoland is the only spot in the German Bight where brachiopods have been found in the past.

The collections of the Museum für Naturkunde Berlin hold three lots of extant brachiopods collected at or around Helgoland. Two of which (ZMB Bra 1325 and 1333) contain a total of three specimens of the zeilleroid *Macandrevia cranium* preserved in alcohol, a species mainly known from subarctic waters [1] but also present off Scotland, the Orkneys and Hebrides [2]. The third sample (ZMB Bra 1396) contains three dry dorsal valves of the craniid *Novocrania anomala*. A lead pencil note on the back of the label by J.-G. Helmcke speculates that this sample was almost certainly dredged in the Helgoland trench ("Tiefe Rinne", Fig 1A), a marine biodiversity hot spot in 50m depth south-west of the island. Helmcke assumes that the shells were drifters originating from the British coast. All three lots were collected in 1888 almost certainly by Arthur Krause, a teacher from Berlin, who was mainly known for travelling East Siberia and Alaska together with his elder brother Aurel Krause in 1881–1882 resulting in a well-received publication on the indigenous Alaskan *Tlingit* tribe.

Interestingly, there is a second record of *N. anomala* from Helgoland. It is described in the reports of the German Pommerania expeditions, a large undertaking supported by the Prussian Ministry of Agriculture to explore the German North Sea and Baltic with regard to oceanography, biology and fisheries. In volume two on the summer expeditions in 1872 and 1873, the list of mollusc samples [3] mentions *N. anomala* from the north harbour of Helgoland on the Eastern side of the island (according to Wilhelm Dunker, a mollusc specialist and professor of zoology at the university of Marburg), and from "Nathurn" (Fig 1B), the Frisian name of the solitary sandstone pinnacle in the North-West of the island, also known as "Lange Anna" (according to Eduard von Martens, then curator of the mollusc collection at the Museum für Naturkunde Berlin). The Pommerania samples are preserved in the Zoological Museum Kiel. The only brachiopod shells found there recently are from Norway, the Helgoland samples seem to be missing (D. Brandis, pers. comm.). Another possibility is that the two records of *N. anomala* from Helgoland by Dunker and v. Martens were not actually collected during the Pommerania expedition but only cited by Metzger in his list of samples as an additional information. Unfortunately, the text in the list is rather ambiguous in this regard.

Throughout the 20th century and up until today not a single observation of a living brachiopod on Helgoland or anywhere else in the German Bight has been published. Due to the oceanographic conditions in the North Sea basin, the mainly soft nature of the sediments and the isolated position of Helgoland, extant brachiopods if at all present seem to be very rare in the German Bight. Additionally, those few sightings recorded in collections or cruise reports are almost certainly the product of drift larvae, which managed to stay alive riding the currents from the British isles or the channel and to establish themselves on rocks or shell beds around Helgoland.

Here we report a recent finding of the smallest extant brachiopod *Gwynia capsula* attached to bryozoans colonizing a dead shell of the bivalve *Modiolus modiolus* from the Helgoland trench. With regard to *Gwynia*'s reproductive biology this observation may be the first indication of a stable population of extant brachiopods in German waters.

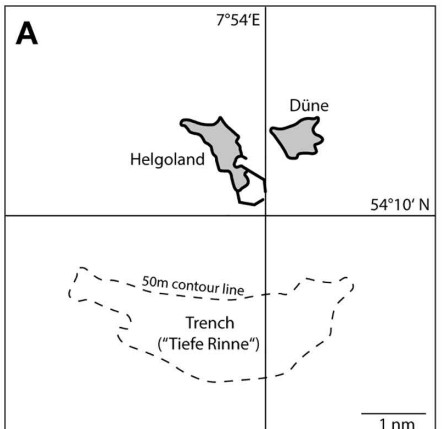

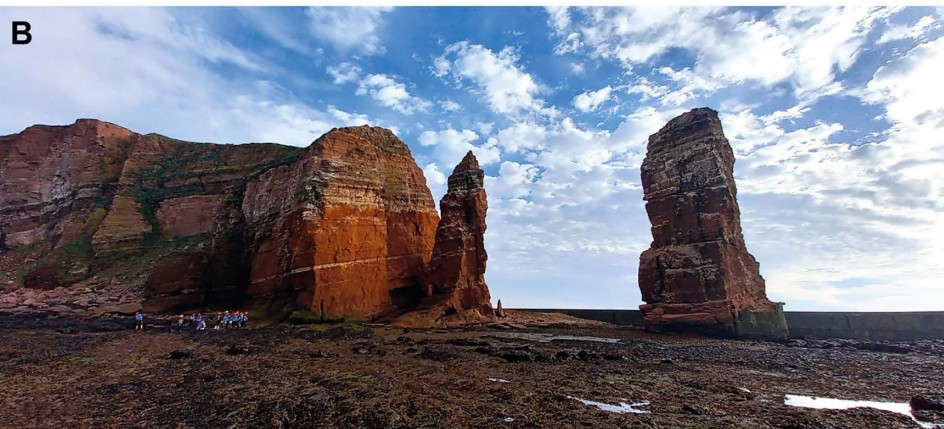

**Fig. 1. Helgoland. A.** Map of the island showing the position of the Helgoland trench ("Tiefe Rinne"). Map re-drawn from [OpenStreetMap](). **B.** North end of main island with the isolated pinnacle "Lange Anna" or Frisian "Nathurn" to the right.

## Materials and methods

### Collection of material

A single living specimen of *Gwynia capsula* (Jeffreys, 1859) was found while browsing through a collection of dead *Modiolus* and *Ostrea* shells dredged by the MS Aaade on 25th September 2023 in the Helgoland Trench ("Tiefe Rinne", [Fig 1A]). As these shells were collected as course material, there is no record of GPS and depth details, but we can assume that the samples were dredged from about 50m depth. The specimen of *G. capsula* settled on the fringe of a circular bryozoan colony of *Escharella immersa* (Fleming, 1828) which itself was growing on the inside of a dead and worn shell of *Modiolus modiolus* (Linnaeus, 1758), a rather large relative of the common black mussel. The coarse sediment at the Helgoland trench mainly consists of these dead bivalve shells of *M. modiolus* and the extinct oyster *Ostrea edulis* and is home to a large variety of marine invertebrates, incl. cnidarians, gastropods, bryozoans, crustaceans, pygnogonids, echinoderms, and tunicates to name only a few.

### Sample preparation

We only discovered the small *G. capsula* specimen due to its slow but regular shell movements. The specimen was left in situ and fixed with 96% ethanol. The whole *Modiolus* shell together with all epibionts incl. the brachiopod was then air dried.

The specimen was examined in a ZEISS Evo LS10 scanning electron microscope without sputter coating using the environmental vacuum mode and a BSE detector. Additionally, part of the *Modiolus* shell including the bryozoan colony and the brachiopod was prepared for µCT imaging at the Museum's µCT facility. Here, the specimen was subjected to micro-tomographic analysis using a Phoenix nanotom X-ray|s tube (Waygate Technologies, Wunstorf, Germany; RRID:SCR_022582) at 90kV and 100µA, generating 2000 projections with 750ms per scan, which results in an effective voxel size of 2.3µm. The cone beam reconstruction was performed using the datos|x 2.2 reconstruction software (Waygate Technologies, Wunstorf, Germany) and the data were visualized in VG Studio Max 3.5 (Volume Graphics GmbH, Heidelberg Germany). For publication images were finally processed with Adobe CC 2023.

The specimen of *G. capsula* is registered in the brachiopod collection of the Museum für Naturkunde, Berlin under the accession number ZMB Bra 2496.

## Ethics statement

The study was conducted in Germany and involved marine invertebrates that are not protected or endangered species. According to the German Animal Welfare Act (*Tierschutzgesetz*), ethical approval is required only for experiments involving vertebrates and cephalopods; therefore, no ethical approval was required for this study. The animals were collected and provided by the host institute in accordance with local and national regulations. All procedures were carried out in a manner designed to minimize stress and harm to the organisms. Animals were euthanized using accepted methods appropriate for marine invertebrates.

## Results

The specimen of *Gwynia capsula* was found attached to the edge of a circular colony of the bryozoan *Escharella immersa*. The bryozoan colony itself together with two tubes of the sedentary polychaete *Spirobranchus triqueter* was attached to the inside of a dead shell of the bivalve *Modiolus modiolus* (Fig 2A). The brachiopod shell (from umbo to frontal margin) is about 500μm long and 400 μm wide (Fig 2B). This perfectly falls into the size range of *G. capsula*. Its short pedicle anchors in the orifice of an *Escharella* cystid (Fig 2C). This is rather unusual, since bryozoans seek to keep the surfaces of their colonies clean in order not to disturb their highly effective filter mechanism. It is likely that the individual bryozoan zooid was already dead when the brachiopod larva attached to it. The surface of the *Gwynia* shell is rather smooth with faint concentric growth lines and a very fine but irregular radial sculpture (Fig 2C). No chaetae can be seen protruding from the shell commissure. μCT-images offer a view inside the brachiopod shell whithout opening it physically. The species-specific submarginal ridges, which support the trocholophus lophophore, are obvious (Fig 2D-F). They are blade-like extensions of the dorsal valve's floor and form an erect curvature from the hinge line towards the front of the shell tapering at about two-thirds of the total shell length (Fig 2E, F). The punctae of *G. capsula* are relatively large (Fig 2B, E). In total, about 60 shells of *Modiolus modiolus* and *Ostrea edulis* were scanned for additional individuals of *Gwynia*, but the described specimen remained the only one.

## Discussion

### Life style

*Gwynia capsula* has been found in many places along the European Atlantic coast, in the Mediterranean and in the Caribbean [1]. Jeffreys [4] already mentions shells from localities such as Ireland, Normandy and the Mediterranean as a basis for his original description of the species. Due to its minute size of around 1 mm *G. capsula* is certainly often overlooked in samples from the field and we can assume that it is much more widespread than so far described in the literature. This has also been suggested by Hansen [5] mentioning the absence of living *G. capsula* from Norwegian waters despite a single fossil specimen described by Sars [6] as a variety of the species from Holocene shell deposits on Kirkøya at the Oslo Fjord, Norway. Unfortunately, Sars' specimen is missing from the collections of the Natural History Museum in Oslo, thus its taxonomic identification cannot be reviewed [5].

With regard to the preferred substrate *Gwynia* seems to be variable [7–9], but many observations describe this smallest extant brachiopod to occur in shell gravel attached to coarse sediment particles such as bivalve shells or broken pieces thereof or to the inside of abandoned calcified worm tubes of, e.g., the serpulid *Spirobranchus triqueter* (e.g., [10] and pers. observ.). This infaunal life style together with its minute size make *G. capsula* the only meiofaunal brachiopod species [11–13]. It occurs in shallow water with max. depth recordings of about 50 m [8]. However, *Gwynia* was described from 100m depth dredged at the Western Approaches, UK, but these were dead shells which may have been transported from shallower waters [14]. According to Logan and coauthors [8] depth recordings of 882m or even 4060m of *G. capsula* from the eastern Atlantic [15] are doubtful (see also [16]). However, its sister species *G. macrodentata* has been described from more than 1800m depth from a seamount east of New Zealand [17].

 

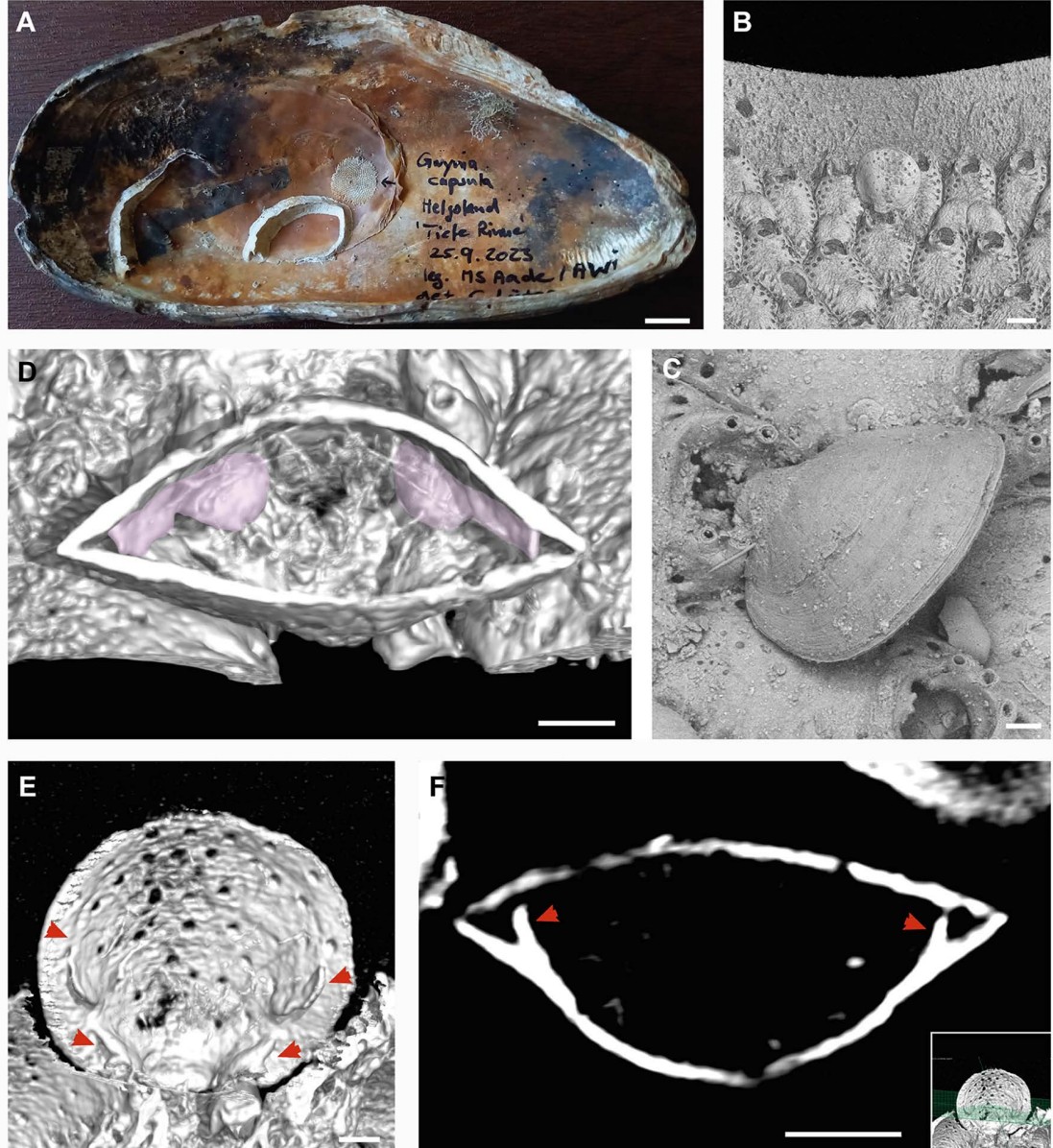

**Fig. 2. *Gwynia capsula* (Jeffreys, 1859) from the Helgoland trench. A.** Dead shell of *Modiolus modiolus* (L.) with several epibionts like calcitic tubes of the serpulid polychaete *Spirobranchus triqueter* (L.) and the circular colony of the bryozoan *Escharella immersa* (Fleming, 1828). The brachiopod is attached to the right margin of the bryozoan colony (black handwritten arrow). **B.** μCT scan of *G. capsula* showing its attachment to a dead *Escharella* cystid. **C.** Same specimen (environmental SEM scan without gold coating, backscatter detector) showing shell surface with no obvious surface details but faint growth lines and a very fine and irregular radial structure. **D.** μCT scan as a 3D-reconstruction of a mid-valve cross section. Submarginal ridges highlighted in pink. **E.** μCT scan of the dorsal valve (optical longitudinal section) showing the submarginal ridges (red arrows) and the relatively large punctae. **F.** Optical cross section of the μCT scan image series showing the height of the submarginal ridges in the dorsal valve (red arrows). Insert indicates the section plane. Scale bars: A: 8 mm, B: 200 μm, C: 50 μm, D: 75 μm, E: 50 μm, F: 80 μm.

## Reproductive biology and dispersal abilities

Being a tiny brachiopod, *Gwynia capsula* does not have the capacity for mass production of eggs or larvae. This is in line with observations on other small brachiopod species, which in many cases tend to retain their few fertilized eggs and brood their embryos inside the shell until they reach a mature stage for subsequent release from the parent and immediate metamorphosis (e.g., Megathyridoidea which *Gwynia* was long thought to be part of) [18], but see discussion in [8,19]. Swedmark [10] was first to describe the reproduction of *G. capsula*. He observed that the female parent only carries two developing eggs/larvae at a time (see also [13]). Since both larvae are not released before they have reached the so-called 3-lobed stage [10], and because these larvae are lecithotrophic they will have to attach to a suitable substrate and metamorphose into a feeding juvenile almost immediately after "hatching" from the mother's brood pouch. Thus, the species' dispersal distance per generation must be very small.

What does that mean for the *Gwynia* specimen found on Helgoland? Although *G. capsula* does not occur in high densities, one can assume that the single specimen found is not the only one inhabiting the shell gravel of the Helgoland trench. A recent long-distance dispersal of larvae of *G. capsula* from either the British coasts or the continental coast along the channel is unlikely regarding the species' reproductive biology. It is rather assumed that a stable population of *G. capsula* inhabits the suitable coarse sediments in the Helgoland trench, but has escaped attention so far. This is unusual though, because the biodiverse fauna of the Helgoland trench has been attracting the interest of marine biologists for a long time [20,21] and is monitored on a regular basis by the Helgoland branch of the Alfred-Wegener-Institute. But how did they get there in the first place?

## Did humans facilitate dispersal of *Gwynia capsula*?

As mentioned in the introduction, the occurrence of dead shells of the inarticulated brachiopod *Novocrania anomala* in waters around Helgoland was attributed to "potential drift of shells" from the British coast. What if this "drift" was anthropogenic? Coarse sediments from British shores containing brachiopod shells or even small living brachiopods could have been used as ballast in ships travelling between Helgoland and Great Britain. Since Helgoland was a British crown colony between 1809 and 1890 there was frequent traffic between the island and the British mainland. After Napoleon had banned the British Empire and Ireland through a continental blockade, Helgoland was used as a stepping stone to smuggle goods from the Empire to the continent and vice versa [22]. Hundreds of vessels per day called at the small port of Helgoland, bearing the chance that sediments were at least sometimes used to stabilize ships with unbalanced loads and were unloaded just before reaching the island. With regard to *G. capsula* this may mean that an initial cohort of this species was literally "sown" in the early 19th century off the Helgoland coast and has afterwards established itself as a stable population of the World's smallest extant brachiopod in the German Bight. However, based on the Holocene fossil occurrence of *G. capsula* in Norwegian waters (see above), a wider distribution in the Atlantic region including the North Sea through natural dispersal over longer periods of time (> 1 mya) cannot be ruled out.

## Acknowledgments

Thanks are due to the Helgoland branch of the Alfred-Wegener-Institute for collecting the shell gravel at the Helgoland trench and to Uwe Nettelmann in particular for preserving the samples for us from a previous field course he supervised. Kristin Mahlow-Tillack, Museum für Naturkunde, has helped with the µCT scans. We also thank Joachim Scholz, Naturmuseum Senckenberg, for identification of the bryozoan colony our brachiopod *G. capsula* is attached to.

## Author contributions

**Conceptualization:** Carsten Lüter.

**Investigation:** Carsten Lüter, Anke Sänger, Fabia Wagner.

**Methodology:** Carsten Lüter, Anke Sänger, Fabia Wagner.

**Project administration:** Carsten Lüter.

**Supervision:** Carsten Lüter.

**Writing – original draft:** Carsten Lüter.

**Writing – review & editing:** Carsten Lüter, Anke Sänger, Fabia Wagner.

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
