## [Decision Letter · Decision Letter 0]

17 Dec 2025

Dear Prof. Lüter,

Thank you for submitting your manuscript to PLOS ONE. After careful consideration, we feel that it has merit but does not fully meet PLOS ONE’s publication criteria as it currently stands. Therefore, we invite you to submit a revised version of the manuscript that addresses the points raised during the review process.

We look forward to receiving your revised manuscript.

Kind regards,

Mikołaj K. Zapalski, Ph. D., D. Sc., Professor

Academic Editor

PLOS One

3. We note that Figures 1 and 2 in your submission contain copyrighted images. All PLOS content is published under the Creative Commons Attribution License (CC BY 4.0), which means that the manuscript, images, and Supporting Information files will be freely available online, and any third party is permitted to access, download, copy, distribute, and use these materials in any way, even commercially, with proper attribution. For more information, see our copyright guidelines: http://journals.plos.org/plosone/s/licenses-and-copyright.

1. You may seek permission from the original copyright holder of Figures 1 and 2 to publish the content specifically under the CC BY 4.0 license.

Additional Editor Comments:

Dear Professor Lüter,

we have now received three reviews of your manuscript. All three reviewers were very positive about your work and I will be very happy to accept it after minor revision.

When introducing small amendments requested by reviewers, please indicate also the repository of the specimen and its inventory number (in the "Materials and Methods" section). You also mention Arthur Krause, his expeditions to Alaska and Siberia and work on Tlinkit tribe - it is not evident for me whether the tribe was from Alaska or Siberia - please clarify this. Actually I am very happy to see a paper that is not purely technical, but adds some historical background - such papers are nowadays very rare.

Yours sincerely,

Mikołaj Zapalski

Reviewers' comments:

Reviewer's Responses to Questions

**Comments to the Author**

1. Is the manuscript technically sound, and do the data support the conclusions?

Reviewer #1: Yes

Reviewer #2: Yes

Reviewer #3: Yes

2. Has the statistical analysis been performed appropriately and rigorously?

Reviewer #1: N/A

Reviewer #2: N/A

Reviewer #3: N/A

3. Have the authors made all data underlying the findings in their manuscript fully available?

Reviewer #1: Yes

Reviewer #2: Yes

Reviewer #3: Yes

4. Is the manuscript presented in an intelligible fashion and written in standard English?

Reviewer #1: Yes

Reviewer #2: Yes

Reviewer #3: Yes

Reviewer #1: Dear sir

This short paper describes the presence of a single specimen of the tiny brachiopod Gwynia capsula from an unusual locality, the island Helgoland in the German Bight.

The very short free-swimming stage of Gwynia larvae and the difficult ocean currents around Helgoland suggest that the brachiopod was possibly introduced by human activities.

The paper is well written, well argued and of interest to a range of scientific disciplines. I recommend that it be published without revision.

Thank you

Jeffrey Robinson

Reviewer #2: The paper presents a new recent finding of a very small brachiopod Gwynia capsula. This discovery is very interesting, additionally when potential human participation in dispersal of G. capsula seems to be very probable. The manuscript is well written. I recommend the paper for publication. I have only a few comments.

The position Metzger (1875) cited in the text was not found in the References.

Caspers (1939) is cited as 1938 in the text.

Hansen (2024) mentioned that Gwynia capsula was found as a subfossil in the Holocene deposits at the Olsofjord. He suggests that this species may still occur today in southern Norway, only being overlooked. The Authors do not cite that paper.

M.A. Bitner

Reviewer #3: The manuscript presents the results of original research, which, to my knowledge, have not been published elsewhere. Gwynia capsula is currently the smallest known recent brachiopod species, which is difficult to find and identify due to its size and is therefore likely to be significantly underrepresented in the literature, and its distribution range may be considerably larger than currently known. This collection bias may be partly responsible for the fact that it has not yet been found in the German part of the North Sea. At the same time, it is also important to note that the soft substrate prevailing in the area is generally unsuitable for the permanent settlement of brachiopods. The paper describes the small number of brachiopods known from the study area to date and the circumstances of the collection of the material currently available. As only a single brachiopod specimen was found, which should not be physically damaged, its internal morphological characteristics were examined using microCT, and its identification within the species Gwynia capsulata was convincingly confirmed. Since the distribution of the species is known to be very patchy, and taking into account its reproductive biology (the very short lecithotrophic larval stage), the authors suggest that Gwynia may have reached the studied area in the 19th century on commercial ships. Although this is a possibility, I consider natural migration to be also a possibility for two reasons. On the one hand, the current distribution of Gwynia capsula is very poorly known due to the lack of knowledge resulting from its small size, so there may be many unmapped stable populations within the North Sea, meaning that this species does not necessarily have to migrate long distances to the German Bight. Secondly, the paper only discusses the recent distribution of the species and only takes this data into account for possible migration. The paper makes no mention of the very limited but existing fossil occurrence of Gwynia (Norway, Pleistocene). If mapping recent occurrences is difficult, identifying fossil records in screen-washed residues is even more problematic, and therefore the fossil record probably even more underrepresented. If we think about the spread of the species not only in the present, but also on a geological time scale, then despite the short larval stage, migrations over very significant distances may have occurred over millions of years.

**Do you want your identity to be public for this peer review?** For information about this choice, including consent withdrawal, please see our Privacy Policy

Reviewer #1: **Yes:** Jeffrey Robinson

Reviewer #2: No

Reviewer #3: No

---

## [Author Response · Author response to Decision Letter 1]

6 Jan 2026

As has been laid out in the response letter to the reviewers, all comments by the editors and the reviewers have been considered and the text of the manuscript as well as the figures have been changed accordingly. For details see the attached response to reviewers letter (in pdf format).

---

## [Editor Report · Decision Letter 1]

11 Jan 2026

First record of the smallest extant brachiopod Gwynia capsula (Jeffreys, 1859) from the German Bight

PONE-D-25-60587R1

Dear Dr. Lüter,

We’re pleased to inform you that your manuscript has been judged scientifically suitable for publication and will be formally accepted for publication once it meets all outstanding technical requirements.

Kind regards,

Mikołaj K. Zapalski, Ph. D., D. Sc., Professor

Academic Editor

PLOS One

Additional Editor Comments (optional):

Dear Professor Lüter,

thank you very much for submitting the revised version of your manuscript. All requested changes have been introduced and all queries answered, and I am happy to inform you that your manuscript can be published now.

My best regards,

Mikołaj Zapalski
---

## [Editor Report · Acceptance letter]

PONE-D-25-60587R1

PLOS One

Dear Dr. Lüter,

I'm pleased to inform you that your manuscript has been deemed suitable for publication in PLOS One. Congratulations! Your manuscript is now being handed over to our production team.

Kind regards,

on behalf of

Prof. Mikołaj K. Zapalski

Academic Editor

PLOS One